# Evaluation of High Intracranial Plaque Prevalence in Type 2 Diabetes Using Vessel Wall Imaging on 7 T Magnetic Resonance Imaging

**DOI:** 10.3390/brainsci13020217

**Published:** 2023-01-28

**Authors:** Masaharu Shozushima, Futoshi Mori, Satoshi Yashiro, Yusuke Todate, Tomoyasu Oda, Kan Nagasawa, Yutaka Hasegawa, Noriko Takebe, Makoto Sasaki, Yasushi Ishigaki

**Affiliations:** 1Division of Diabetes, Metabolism and Endocrinology, Department of Internal Medicine, Iwate Medical University, Yahaba 028-3695, Japan; 2Division of Ultra-high Field MRI, Institute for Biomedical Sciences, Iwate Medical University, Yahaba 028-3694, Japan

**Keywords:** vessel wall imaging, intracranial plaques, diabetes, high resolution magnetic resonance imaging

## Abstract

Background: While type 2 diabetes (T2D) is a major risk for ischemic stroke, the associated vessel wall characteristics remain essentially unknown. This study aimed to clarify intracranial vascular changes on three-dimensional vessel wall imaging (3D-VWI) using fast spin echo by employing 7Tesla (7T) magnetic resonance imaging (MRI) in T2D patients without advanced atherosclerosis as compared to healthy controls. Methods: In 48 T2D patients and 35 healthy controls, the prevalence of cerebral small vessel diseases and intracranial plaques were evaluated by 3D-VWI with 7T MRI. Results: The prevalence rate of lacunar infarction was significantly higher in T2D than in controls (*n* = 8 in T2D vs. *n* = 0 in control, *p* = 0.011). The mean number of intracranial plaques in both anterior and posterior circulation of each subject was significantly larger in T2D than in controls (2.23 in T2D vs. 0.94 in control, *p* < 0.01). In T2D patients, gender was associated with the presence of intracranial plaques. Conclusion: This is the first study to demonstrate the high prevalence of intracranial plaque in T2D patients with neither confirmed atherosclerotic disease nor symptoms by performing intracranial 3D-VWI employing 7TMRI. Investigation of intracranial VWI with 7T MRI is expected to provide novel insights allowing early intensive risk management for prevention of ischemic stroke in T2D patients.

## 1. Introduction

Type 2 diabetes (T2D) is a major risk factor for ischemic stroke, which often results in physical impairment and cognitive dysfunction [1]. East Asian populations especially, as compared to Caucasians, have been characterized as having high morbidity and mortality due to cerebral stroke [2]. In Japan, the risk of cerebral infarction after adjustment for multiple potential factors was found to be approximately two to four times higher for subjects with T2D than those with normal glucose tolerance [3]. A large Asian cohort study showed T2D to be a significant risk factor for ischemic strokes of three major subtypes, i.e., lacunar, large-artery occlusive, and embolic infarctions [4]. While intensive investigations have revealed the pathogenic factors associated with cerebral atherosclerosis in T2D, including intra-arterial stenosis or fibrous cap rupture of atheroma accompanied by hyperglycemia as well as insulin resistance, the mechanisms have yet to be fully elucidated. Brain imaging studies may facilitate clarifying the mechanisms underlying vascular damage, but unexpectedly, the association of T2D with the prevalence of cerebral small vessel disease (SVD), such as deep white matter hyperintensities (DWMH) and cerebral microbleeds (CMBs) [5], is not yet understood. Thus, more precise investigations of the cerebral vascular state in T2D patients are required.

Aiming to evaluate intracranial vascular changes in terms of lumen caliber, magnetic resonance angiography (MRA), computed tomography angiography (CTA), and digital subtraction angiography (DSA) have been used, but accurately visualizing morphological changes in cerebral vessel walls is found to be difficult. Intracranial vascular imaging advancements in recent years have been achieved by applying high-resolution vessel wall imaging (VWI) employing magnetic resonance imaging (MRI), providing useful information on plaque characteristics including morphology and components [6,7,8,9]. Concurrently with improvements in imaging techniques and protocols [9], VWI investigations have become increasingly widely used in various disease types, including T2D. In T2D patients with cerebrovascular symptoms, poor glycemic control reportedly has a greater impact on the disease burden and the vulnerability of intracranial atherosclerotic plaques [10]. In addition, a study recruiting patients with acute ischemic stroke revealed an association between having diabetes and intracranial plaque number as well as high hemoglobin A1c (HbA1c) and stronger plaque enhancement on imaging [11].

Furthermore, the development of ultra-high-field 7.0 Tesla (7T) MRI, which provides an increased signal-to-noise ratio (SNR) of the inflow signal at a high spatial resolution [12], demonstrates more detail regarding vascular conditions [13,14]. While the most current intracranial VWI studies have been performed at a 3-Tesla field strength, a few VWI investigations using 7T MRI equipment have recently been examined to allow for high-resolution imaging [15,16]. Zwartbol et al. performed VWI employing 7T MRI on patients with a history of vascular disease and showed a significant association of intracranial atherosclerosis with presence of diabetes [17]. However, all of patients enrolled in this study had a past history of severe atherosclerosis, and only 19% had diabetes. Taking the findings of prior studies together, most of the enrolled patients had either symptoms or at least one comorbidity of cerebrovascular disease such that wall vessel lesions in T2D patients without advanced cerebral atherosclerosis remain a largely unexplored topic.

Therefore, in order to determine whether intracranial vascular changes are associated with diabetes and if so, their nature and severity, we designed this cross-sectional study using high resolution VWI at 7T MRI in T2D patients without advanced atherosclerosis.

## 2. Materials and Methods

### 2.1. Study Subjects

The study subjects were 48 T2D patients admitted to Iwate Medical University Hospital during the period from November 2014 to May 2021. T2D was defined as taking glucose-lowering medications or HbA1c ≥ 6.5% or fasting blood glucose ≥ 126 mg/dL, on the basis of the diagnostic criteria proposed by the Japan Diabetes Society [18]. Hypertension was defined as systolic blood pressure (sBP) ≥ 140 mmHg and/or diastolic blood pressure (dBP) ≥ 90 mmHg and/or taking antihypertensive medications. Dyslipidemia was defined as low-density lipoprotein cholesterol ≥ 140 mg/dL and/or triglycerides ≥ 150 mg/dL and/or high-density lipoprotein cholesterol (HDL-C) < 40 mg/dL and/or taking antihyperlipidemic medications. Diabetic retinopathy was diagnosed by ophthalmologists based on the international clinical diabetic retinopathy scales. Diabetic nephropathy was defined as urine albumin excretion (UAE) ≥ 30 mg/g creatinine [18]. Diabetic neuropathy was defined as the presence of two of the following three findings: typical subjective symptoms of symmetrical distal neuropathy, bilaterally decreased Achilles tendon reflexes, or an inability to sense vibration [18].

Thirty-five metabolically healthy volunteers were enrolled as controls. The exclusion criteria applied to assure “metabolically healthy” status were the absence of hyperglycemia, high blood pressure, and lipid profile abnormalities at the most recent medical check-up and no past history of diabetes and/or hypertension and/or dyslipidemia.

None of the study subjects had any history of either coronary heart disease, cerebrovascular disease, or peripheral artery disease. Written informed consent was obtained from all participants. This study was approved by the Institutional Review Board of Iwate Medical University (MH2019-156). The study was conducted according to the Declaration of Helsinki.

### 2.2. MR Protocols

We used a 7T MRI scanner (Discovery MR950; GE Healthcare, Milwaukee, WI, USA) with quadrature transmission and a 32-channel receive head coil. Sagittal T1-weighted (T1W) three-dimensional (3D) fast spin echo (FSE) VWI [19,20,21] performed by applying the following parameters: TR, 600 ms; TE, 14.4 ms; length of echo train, 8; FA, 90°; FOV, 20 cm; matrix size, 512 × 224; slice thickness, 0.8 mm; reconstructed voxel size, 0.39 × 0.39 × 0.4 mm (after zero-fill interpolation); number of slices, 452; and acquisition time, 13 min 30 s. Furthermore, high-resolution 3D time-of-flight (TOF) MRA [22,23] was acquired using a 3D spoiled gradient recalled echo sequence with the following scanning parameters: TR, 12 ms; TE, 2.4 ms; FA, 12°; FOV 240 mm; matrix size 768 × 384; slice thickness, 0.6 mm; reconstructed voxel size, 0.23 × 0.23 × 0.3 (after zero-fill interpolation); number of slices, 352; and acquisition time, 10 min 26 s. In addition, conventional brain MRI, e.g., T1-weighted, T2*-weighted, and fluid-attenuated inversion recovery (FLAIR) images, were also obtained [24]. We performed 3D-VWI to evaluate vessel wall lesions; 3D TOF MRA to evaluate vessel lumen; T1-weighted and FLAIR images to evaluate lacunar, PVH, DWMH, and brain atrophy; and T2*-weighted to evaluate CMBs.

### 2.3. Data Analysis

Three authors (M.Sh., S.Y., and Y.T.) blinded to the subjects’ clinical and demographic characteristics, i.e., age, sex, body weight, body mass index (BMI), sBP, dBP, smoking history, HbA1c, liver enzymes, and lipid profiles, analyzed the data. Moreover, two authors (M.Sh. and F.M.) performed construction of the curved planar reformation (CPR) images, obtained by 3D-VWI, including bilateral imaging of the internal carotid artery (ICA), middle cerebral artery (MCA), vertebral artery (VA), and basilar artery (BA), using a 3D workstation (Ziostation 2; Ziosoft Inc., Tokyo, Japan). 

3D-VWI images were reformatted to achieve short-axis multi-planar reconstruction (MPR) and long-axis CPR to delineate vascular wall properties in the intracranial arteries. The images thus obtained were visually evaluated. The short-axis MPR of the horizontal portions of the intracranial arteries (ICA, MCA, VA, and BA) were generated with 1.0 mm intervals and FOV 80 × 80 mm. Moreover, long-axis CPR images were created every 5º for 360º. By using these images, the presence or absence of plaque, the shapes of plaques, and the signal intensity of plaques in the intracranial arteries could be compared between T2D and control subjects. According to previous investigations, determining cut-off values for VWI with MRI correspond to relevant histological findings of intraplaque components [25].

A board-certified senior radiologist (M.Sa., with more than 20 years of experience) blinded to the clinical status of the patients visually evaluated all images twice each for the presence of any abnormalities. This radiologist concurrently determined narrowing or interruption of the ICA, MCA, VA, and BA, as indicated by a decrease in signal intensity on the MRA-MIP images due to reduced blood flow. Conventional brain MRI scans were also obtained in order to assess DWMH and periventricular hyperintensity (PVH), applying the Fazekas grade, lacunar infarctions, brain atrophy, and microbleeds [26].

### 2.4. Statistical Analysis

Quantitative data are presented as the mean with standard deviation (SD). Statistical analyses were conducted employing the Student’s *t*-test with data showing a normal distribution, while the Mann–Whitney U test was used for those showing a non-normal distribution. Chi-square test or Fisher’s exact test was used to determine associations between two categorical variables. The significance level was set at *p* < 0.05. All statistical analyses were performed using SPSS version 26 (SPSS Japan Inc. Tokyo, Japan).

## 3. Results

Among those receiving MR examinations, one of the control subjects insisted upon cessation of the scanning procedures due to claustrophobia. The remaining 35 controls and all 48 T2D patients were eligible for further analyses. 

The clinical characteristics of the enrolled subjects were shown in Table 1. The average HbA1c was 9.5% in T2D patients. Age and proportion of males were similar, but body weight, BMI, sBP, dBP, proportion of current or former smokers, and triglyceride levels were higher in the T2D than in the control group. In contrast, HDL-C values were lower in the patients with T2D than in the controls. In the T2D group, 43.8% of patients had the comorbidity of hypertension, and 66.7% of these patients also had dyslipidemia.

The conventional brain MRI scans revealed that the prevalence of lacunar infarction was significantly high in T2D compared with controls (*n* = 8 in T2D vs. *n* = 0 in control, *p* = 0.011). In contrast, no significant difference was found in the proportion of CMBs, DWMH, PVH, and brain atrophy between the two groups (Table 2). Next, apparent brain damage and large-vessel abnormalities were analyzed using 3D-VWI with 7TMRI (Figure 1). On high-resolution 3D-VWI, eccentric plaques were identified significantly more often in T2D patients than in controls. In both anterior and posterior circulation, the mean number of plaques in T2D patients was 2.23, whereas that of controls was 0.94, suggesting the prevalence of intracranial plaque to be significantly higher in the former (*p* < 0.01). Similarly, in the anterior circulation, the mean number of plaques in T2D patients was larger than in controls (1.52 vs. 0.51, *p* < 0.01), while no marked difference was detected in the posterior circulation (T2D 0.71 vs. controls 0.43, *p* = 0.069) (Table 3). 

In order to clarify the factors influencing intracranial plaque formation, T2D patients were divided into two groups according to the presence of plaques in both anterior and posterior circulation, and differences in clinical parameters were compared. The only factor associated with the presence of plaques in T2D patients was gender (Table 4). Prevalence of comorbidities, i.e., hypertension and/or dyslipidemia, did not differ between the two groups.

## 4. Discussion

To our knowledge, this is the first study to demonstrate an increased number of intracranial plaques using high-resolution 3D-VWI with 7T MRI in T2D patients without apparent atherosclerotic disease. While the prevalence of SVD in those with T2D had yet to be fully clarified, these results suggested that a large proportion of asymptomatic T2D patients may have early-stage cerebral atherosclerosis. Investigation of intracranial VWI with 7T MRI is expected to provide novel insights for early initiation of intensive risk management aimed at preventing ischemic stroke in patients with T2D.

Since several previous studies focusing on vascular wall changes were conducted in patients with comorbidities, either ischemic stroke or transient ischemic attacks, the vessel wall characteristics of asymptomatic subjects are largely unknown. Therefore, we enrolled subjects who had no history of either cerebrovascular diseases or coronary heart disease as well as being free of peripheral artery disease, and we examined whether they had arterial anomalies or gross vascular lumen abnormalities using MRA. Even among seemingly low-risk subjects, a high prevalence of intracranial plaques in T2D patients without atherosclerotic disease confirmed the potential risk for cerebrovascular disease associated with diabetes. These results suggested that early detection of intracranial changes using 3D-VWI might be useful for evaluating the cerebral infarct risk as well as the initiation of intensive treatments for risk factors in asymptomatic T2D patients. 

Intracranial atherosclerosis is regarded as a major cause of ischemic stroke and transient ischemic attack development. Conventional vascular imaging, such as CTA, DSA, and MRA, demonstrated lumen caliber changes [27] but with limited efficacy due mainly to uncertainty in the detection of non-specific signals. The recent development of MRI sequences allows for sensitive detection of vessel wall changes, including those that have not yet caused luminal narrowing, as well as investigation of the underlying pathology in vivo [28]. It is highly likely that identification of plaque characteristics, for instance, intra-plaque hemorrhage, lipid content, and so on, leading to intracranial atherosclerotic disease, will become possible with this innovation [29]. Additionally, in order to optimize the conditions for the assessment of vessel wall changes, we constructed CPR images from all 3D-VWI, including the ICA, MCA, VA, and BA obtained from each subject. Because of the tortuous anatomy of intracranial arteries, an arduous process is required for accurate visualization of 3D-VWI based on stretching these tortuous courses through the application of technical conditions allowed by the current hardware and software environments [30]. Although these burdensome tasks would be hard to practice in routine examinations, the construction of large numbers of CPR images enabled us to determine the number of plaques in major intracranial arteries, leading to the recognition of early cerebral atherosclerosis risk in the asymptomatic T2D patients in this study. 

Another strength of this study is imaging analyses using high-resolution 7T MRI. The advantages of a high-magnetic field include the increased SNR and contrast-to-noise ratios, which can be exploited for imaging at a higher spatial resolution, thereby allowing clear visualization of distal small arterial branches and thin vessel walls by suppressing the background signal [22,23]. Zhu et al. refined high-resolution 3D MRI techniques for intracranial VWI at both 3- and 7T in patients with intracranial artery disease and concluded that the latter provided better image quality and improved confidence in diagnosis [31]. Similarly, a study with asymptomatic elderly volunteers showed visibility of intracranial VWI was equal to or significantly better at 7T than at 3T MRI [32]. In accordance with previous comparative studies, intracranial VWI at 7T MRI has a high capability for visualizing the vascular condition in detail, suggesting that the findings obtained in this study are both highly reliable and informative. While 7T MRI is not as yet being used clinically, a growing body of research might allow the utility of VWI with ultra-high-field MRI for the investigation of cerebral vascular complications in high-risk subjects such as T2D.

The findings obtained from this 7T MRI study, which can be summarized as a high prevalence of intracranial plaque in T2D patients, are novel but apparently obvious, being consistent with those of previous 3T MRI studies [10,11]. Therefore, we performed comparative analyses aimed at determining risk factors for intracranial arterial plaque prevalence in T2D. Unexpectedly, only the male gender was identified as a factor related to plaque prevalence. Several studies have shown HbA1c values were strongly associated with multiple intracranial stenoses as evaluated by MRA, reflecting the severity of intracranial atherosclerosis [33]. In fact, prolonged hyperglycemia is regarded as a major risk factor for atherosclerosis development as well as a microvascular disease [34] due to the accumulation of advanced glycation end-products (AGE) [35], activation of the post-receptor for AGE pathway, and enhancement of the polyol pathway in vascular endothelial cells [36]. However, temporal values of HbA1c do not always reflect long-term glycemic control. Since most of the T2D patients enrolled in this study were compelled to admit for treatment of poor glycemic control, their HbA1c values at the time of investigation may have been at nearly the peak when the MRI examinations were conducted. Recent increases in HbA1c values, which may not actually have reflected long-term poor glycemic control in some of the enrolled patients, might have resulted in the loss of a statistically significant relationship between the presence of plaque and HbA1c values. In addition, age and the presence of hypertension were not related to the presence of intracranial plaque. We cannot precisely explain these unexpected results. In previous reports, neither the presence of hypertension nor current blood pressure was reported to be a significant factor for either the existence or the vulnerability of intracranial plaque in diabetic subjects based on high-resolution MRI investigation [10,11,33], which is consistent with the findings obtained in this study. In our T2D patients with intracranial plaque, the proportions with elevated blood pressure and being administered calcium channel blockers both tended to be high, suggesting that they might have hard-to-treat hypertension requiring multiple medications. Further large-scale studies are required to elucidate the influences of both aging and hypertension on cerebral vessel wall changes in asymptomatic T2D patients. 

The major limitation of this study is its cross-sectional design, raising the possibility that our results show only associations. The relationship between the presence of intracranial plaques and the development of SVD in T2D awaits confirmation in a longitudinal study. Second, 43.8% of our T2D patients had hypertension, and 66.7% of them had dyslipidemia. Thus, the findings related to intracranial plaque cannot be assumed to reflect a direct effect of diabetes itself. Recruiting T2D patients without comorbidities might resolve this issue but would be a highly challenging endeavor. Third, contrast enhancement of intracranial arterial walls reportedly suggests atherosclerotic changes [6,25]. However, we did not obtain postcontrast images in this study, mainly due to both ethical considerations and the burden on patients, i.e., invasiveness. Fourth, quantitative criteria for intracranial plaque characteristics have not yet been established. Fifth, evaluations of VWI by a single board-certified senior radiologist might have led to detection bias of intracranial plaque. Sixth, plaque presence in the circle of Willis, including the anterior cerebral artery and posterior cerebral artery, was unexamined in this study. Seventh, although vessel wall plaques are generally known to exist at the bifurcation or curved portion of the artery, the association of plaque presence with vessel morphology has not been considered. Finally, our sample size was too small to allow sufficiently powered statistical analysis to be performed. Therefore, the results are preliminary rather than definitive, necessitating further studies to test our findings in a large subject population.

## 5. Conclusions

In conclusion, this study is the first demonstration of the high prevalence of intracranial plaque in subjects with T2D by employing intracranial 3D-VWI obtained with 7TMRI. Our data can be applied to elucidating intracranial vessel wall lesions associated with hyperglycemia and also raise the possibility of 3D-VWI being used in clinical practice in the future.

## Figures and Tables

**Figure 1 brainsci-13-00217-f001:**
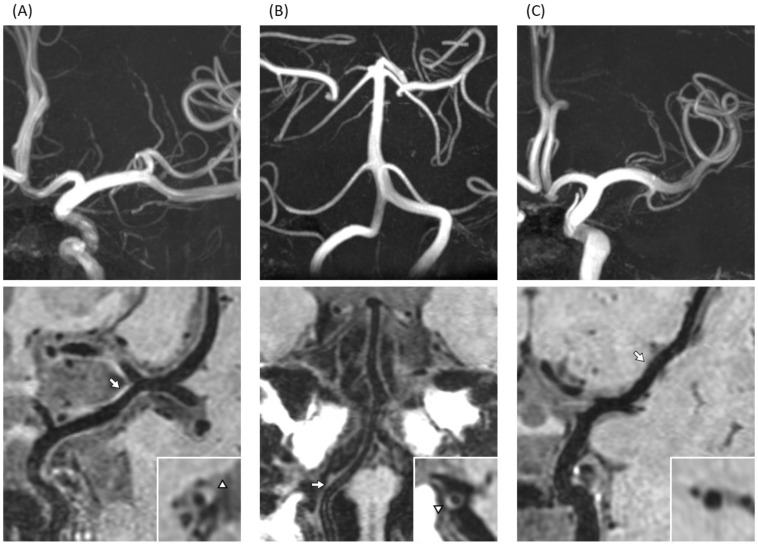
T1-weighted three dimensional vessel wall imaging (3D-VWI) in type 2 diabetes mellitus (T2D) patients and a control. (**A**) A 59-year-old man with T2D had wall thickening at left MCA with high signal indicating atherosclerotic plaque, visualized by curved planar reformation (CPR) (arrow) and axial-sectional view (triangle). (**B**) A 56-year-old woman with T2D showed wall thickening at right VA with a high signal indicating atherosclerotic plaque, visualized by CPR (arrow) and axial-sectional view (triangle). (**C**) No plaques were detected in a 58-year-old woman from the control group, visualized by CPR (arrow) and axial-sectional view at indicated location by arrow.

**Table 1 brainsci-13-00217-t001:** Clinical characteristics of subjects.

	T2D	Control	*p*-Value
	(*n* = 48)	(*n* = 35)	
Age (years)	53.2 ± 6.3	50.7 ± 5.3	0.067
Male (%)	64.6	54.3	0.344
Body weight (kg)	71.7 ± 11.9	61.6 ± 9.4	<0.01
Body mass index (kg/m^2^)	26.1 ± 4.2	22.4 ± 2.3	<0.01
Systolic blood pressure (mmHg)	128.8 ± 14.4	115.9 ± 14.0	<0.01
Diastolic blood pressure (mmHg)	80.4 ± 10.8	73.5 ± 9.5	<0.01
Former or current smoking (%)	50.0	14.3	<0.01
HbA1c (%)	9.5 ± 3.0	5.5 ± 0.3	<0.01
AST (IU/mL)	27.5 ± 24.2	21.4 ± 5.0	0.96
ALT (IU/mL)	33.3 ± 27.4	20.3 ± 9.8	<0.01
γ-GTP (IU/mL)	62.8 ± 105.4	41.9 ± 47.7	0.278
TC (mg/dL)	191.4 ± 49.0	197.9 ± 35.0	0.554
TG (mg/dL)	149.8 ± 77.0	81.7 ± 35.4	<0.01
LDL-C (mg/dL)	114.3 ± 39.1	108.3 ± 23.6	0.42
HDL-C (mg/dL)	51.6 ± 13.8	71.7 ± 15.3	<0.01
Hypertension, *n* (%)	21 (43.8)	0	
RAS inhibitor, *n*	15		
Calcium channel blocker, *n*	9		
Dyslipidemia, *n* (%)	32 (66.7)	0	
Statin, *n*	14		
Fibrate, *n*	6		

Values are presented as means (±SD). Analyzed by Student’s *t*-test or chi-square test. T2D, type 2 diabetes; HbA1c, hemoglobin A1c; AST, aspartate aminotransferase; ALT, alanine aminotransferase; γ-GTP, γ-glutamyl transpeptidase; TC, total cholesterol; TG, triglyceride; LDL-C, low-density lipoprotein-cholesterol; HDL-C, high-density lipoprotein-cholesterol; RAS, renin-angiotensin system.

**Table 2 brainsci-13-00217-t002:** Prevalence of SVD.

	T2D	Control	*p*-Value
Lacunar infarction	8 (16.7%)	0 (0%)	0.011
CMBs	2 (4.2%)	2 (5.7%)	0.565
PVH	2 (4.2%)	0 (0%)	0.311
DWMH	8 (16.6%)	2 (5.7%)	0.119
Brain atrophy	2 (4.2%)	0 (0%)	0.331

Plaque number is presented as numbers (%). Analyzed by Fisher’s exact test. T2D, type 2 diabetes; CMBs, cerebral microbleeds; PVH, periventricular hyperintensity; DWMH, deep white matter hyperintensity.

**Table 3 brainsci-13-00217-t003:** Mean number of intracranial plaques per person.

	T2D	Control	*p*-Value
Anterior and posterior circulation	2.23 ± 0.23	0.94 ± 0.21	<0.01
Anterior circulation	1.52 ± 0.18	0.51 ± 0.12	<0.01
Posterior circulation	0.71 ± 0.12	0.43 ± 0.13	0.133

Plaque number is presented as the mean value ± standard error. Analyzed by Mann–Whitney U test. Anterior Circulation includes internal carotid artery and middle cerebral artery. Posterior Circulation includes vertebral artery and basilar artery. T2D, type 2 diabetes.

**Table 4 brainsci-13-00217-t004:** The risk factors of plaque presence in both anterior and posterior circulation in T2D patients.

	Type 2 Diabetes		*p*-Value
	(+) *n* = 38	(−) *n* = 10	
Age (years)	53.6	51.3	0.310
Male (%)	29 (76.3)	2 (20)	<0.01
sBP (mmHg)	129.6	125.6	0.441
dBP (mmHg)	81.8	74.9	0.060
HbA1c (%)	9.3	10.1	0.572
AST (IU/mL)	29.4	20.1	0.284
ALT (IU/mL)	35.5	25.2	0.297
γ-GTP (IU/mL)	71.2	30.8	0.285
TC (mg/dL)	187.5	206.3	0.285
TG (mg/dL)	155.4	128.6	0.333
LDL-C (mg/dL)	112.6	120.5	0.577
HDL-C (mg/dL)	49.6	59.4	0.430
eGFR (mL/min/1.73m^2^)	74.4	78.8	0.389
Former or current smoking (%)	21(55.2)	3(30.0)	0.155
Hypertension, *n* (%)	17 (42.1)	4 (40.0)	0.542
RAS inhibitor, *n* (%)	12 (31.6)	3 (30.0)	0.602
Calcium channel blocker, *n* (%)	9 (23.7)	0	0.094
Dyslipidemia, *n* (%)	26 (63.2)	6 (60.0)	0.594
Statin, *n* (%)	11 (28.9)	3 (30.0)	0.612
Fibrate, *n* (%)	4 (10.5)	2 (20.0)	0.355
Diabetic neuropathy (%)	22 (57.8)	5 (50.0)	0.654
Diabetic retinopathy (%)	16 (42.1)	5 (50.0)	0.654
Diabetic nephropathy (%)	13 (34.2)	4 (40.0)	0.733
Lacunar infarction (%)	8 (21.1)	0 (0)	0.112
CMBs (%)	2 (5.3)	0 (0)	0.459
PVH (%)	1 (2.6)	1 (10.0)	0.299
DWMH (%)	6 (15.8)	2 (20.0)	0.751
Brain atrophy (%)	2 (5.3)	0 (0)	0.459

Analyzed by Student’s *t*-test or chi-square test or Fisher’s exact test. sBP, systolic blood pressure; dBP, diastolic blood pressure; HbA1c, hemoglobin A1c; AST, aspartate aminotransferase; ALT, alanine aminotransferase; γ-GTP, γ-glutamyl transpeptidase; TC, total cholesterol; TG, triglyceride; LDL-C, low-density lipoprotein-cholesterol; HDL-C, high-density lipoprotein-cholesterol; eGFR, estimated glomerular filtration rate; RAS, renin-angiotensin system; CMBs, cerebral microbleeds; PVH, periventricular hyperintensity; DWMH, deep white matter hyperintensity.

## Data Availability

The data that support the findings of this study are available on request from the corresponding author, Y.I., on reasonable request. The data are not publicly available due to containing information that could compromise the privacy of research participants.

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
