# Peer review of "Evaluation of High Intracranial Plaque Prevalence in Type 2 Diabetes Using Vessel Wall Imaging on 7 T Magnetic Resonance Imaging"

_brainsci, 2023, doi:10.3390/brainsci13020217_

Round 1

Reviewer 1 Report

In this study, the authors analyze the prevalence of intracranial atherosclerotic plaques in patients with type 2 diabetes and compare it to a cohort of healthy patients. They aimed to study the underlying relationship between types and locations of atherosclerotic plaques in T2D and healthy patients. They found that lacunar infarctions were significantly higher in T2D patients than in healthy patients. Additionally, they found that plaques were significantly higher in the anterior circulation of T2D patients as compared to healthy patients but was not so in the posterior circulation. Apart from the gender, none of the other parameters were different between patients with anterior and posterior plaques.

This study has moderate impact in its current state however, very few studies have used high strength MRI scanners to observe the plaque presence. This study can generate higher impact if quantitative morphology is added as well.

Major comments:

  • Instead of classifying them as anterior or posterior, please also run a ANOVA analysis with the plaque presence at different locations in the circle of Willis.
  • With cohorts where expected value is 0 for chi-squared tests (e.g. in table 4, lacunar infarction is 9 and 0), use the fischers exact test.
  • To increaser the impact of the study, is it also possible to compute the general morphology (e.g. the tortuosity, average length, curvature etc.) of the arteries and compare it between the T2D and healthy cohort?

Minor Comments:

  • Typo: Line 117: “internal cerebral artery” I think you mean internal carotid artery.

Please include a table with the distribution of the plaques found at different arteries (ICA, PCOM, MCA, ACOM, BA etc.). 

Reviewer 2 Report

The purpose of the manuscript was to demonstrate the feasibility of using T1-weighted 3D VWI at 7T to detect early signs of intracranial atherosclerosis for the management for prevention of ischemic stroke in T2D patients. The VWI was performed on T2D patients with neither confirmed atherosclerotic disease nor symptoms and closely age-matched controls. Two main criticisms are insufficient detail of the VWI protocol to repeat the experiments and results that were less than compelling. The first can be easily addressed but the latter criticism significantly reduces enthusiasm for the manuscript. It may reflect the fact that detecting early stages of atherosclerosis is no easy task in presymptomatic patients; it may not be possible since plaque formation may represent advanced stage of atherosclerosis. Additional suggestions/comments that need to be addressed are listed below.

Line 16: Authors should state the exact VWI technique used in the abstract.

Lines 101 – 106: Does the sequence start with an inversion? If so, what is the inversion time? Do you implement fat suppression? Briefly describe how 3D FSE suppress blood water proton signal to achieve black blood imaging. What is the turbo factor, i.e. how many k-space lines are collected per TR. For time-of-flight MRA, doesn’t the low flip angle 3D acquisition reduce or remove the flow enhancement effect? It is unclear to me how the 3D sequence will enhance intravascular signal and suppress the static tissue signal at the same time.

Line 111: It would have been interesting to collect hemodynamic information to supplement the structural imaging.

Line 166: Unclear why Fig 1 is referred to as scout images. Typically, scout or localizer images are used to plan and prescribe imaging sequence of interest.

Fig 1: I do not find the images compelling in terms of visualizing plaque formation. Please provide additional images that show vessel wall thickening in T2D. Please explain why hyperintense signal is associated with plaque. Why are there no MRA images? Presumably MRA is unable to detect early signs of lumen narrowing.

Table 3: Please include standard deviation of plaque numbers.

Line 224: Using MRA as a pre-exam was not stated in Methods. Was that the purpose of performing MRA?

Line 231: Replace “VWI technique” with vascular imaging since CTA, DSA and MRA are not VWI techniques.

Line 233-234: Unclear which sequences the authors are referring to because there are several approaches to VWI.

Line 241: accurate visualization of intracranial arteries from VWI is indeed an arduous task for radiologists. This is one of the reasons why VWI is not practical clinically and is not an advantage. Please comment on the burden of reviewing and reformatting so many images. Unlike MRA, maximum intensity projection cannot be applied to identity plaque formation quickly.

Round 2

Reviewer 2 Report

I am still unclear how FSE suppress intravascular signal without using MSDE or DANTE. Authors did not address "Briefly describe how 3D FSE suppress blood water proton signal to achieve black blood imaging." In Ref 23, a completely different sequence was used.  

Hemodynamics refers to flow velocity and MRA does not provide that information.
